# Metabolic Control of the FreeStyle Libre System in the Pediatric Population with Type 1 Diabetes Dependent on Sensor Adherence

**DOI:** 10.3390/jcm11020286

**Published:** 2022-01-06

**Authors:** Isabel Leiva-Gea, Maria F. Martos-Lirio, Ana Gómez-Perea, Ana-Belen Ariza-Jiménez, Leopoldo Tapia-Ceballos, Jose Manuel Jiménez-Hinojosa, Juan Pedro Lopez-Siguero

**Affiliations:** 1Pediatric Endocrinology, Hospital Regional Materno Infantil de Málaga, 29011 Malaga, Spain; isabeleiva@hotmail.com (I.L.-G.); mariaml_huelma@hotmail.com (M.F.M.-L.); gomezpereana@gmail.com (A.G.-P.); leotapiaceb@hotmail.com (L.T.-C.); jmhinojosa@hotmail.com (J.M.J.-H.); lopez.siguero@gmail.com (J.P.L.-S.); 2Multidisciplinary Group on Paediatric Research, Instituto de Investigación Biomédica de Málaga, 29010 Malaga, Spain; 3Pharmacology and Pediatrics Department, University of Malaga, 29016 Malaga, Spain; 4Pediatric Endocrinology, Hospital Universitario Reina Sofia, 14004 Cordoba, Spain

**Keywords:** flash glucose monitoring, sensor scanning, hypoglycemia

## Abstract

Aims: To evaluate the relationship between daily sensor scan rates and changes in HbA1c and hypoglycemia in children. Methods: We enrolled 145 paediatric T1D patients into a prospective, interventional study of the impact of the FreeStyle Libre 1 system on measures of glycemic control. Results: HbA1c was higher at lower scan rates, and decreased as the scan rate increased to 15–20 scans, after which it rose at higher scan rates. An analysis of the change in hypoglycemia, based on the number of daily sensor scans, showed there was a significant correlation between daily scan rates and hypoglycemia. Subjects with higher daily scan rates reduced all levels of hypoglycaemia. Conclusions: HbA1c is higher at lower scan rates, and decreases as scan rate increases. Reductions in hypoglycemia were evident in subjects with higher daily scan rates.

## 1. Introduction

The FreeStyle Libre 1 flash glucose monitoring system (Abbott Diabetes Care, Witney, UK) is an established technology that measures glucose in interstitial fluid (ISF). A sensor worn on the back of the upper arm takes a reading every minute that can be scanned using a hand-held reader or smartphone to receive a current glucose result, along with historic results with a 15-min frequency. The FreeStyle Libre sensors are calibrated in the factory and have a wear time of up to 14 days without the need for the user to perform daily calibration using finger-prick tests [1]. 

The FreeStyle Libre flash glucose monitoring system was proven in the IMPACT and REPLACE studies [2,3] to reduce time in hypoglycemia below 70 mg/dL by 38% (IMPACT) and 43% (REPLACE) over 26 weeks, for adults with type 1 (T1D) or type 2 diabetes (T2D) on insulin, compared with the finger-prick method for the self-monitoring of blood glucose (SMBG). Neither study showed a significant change in HbA1c observed with the flash glucose monitoring compared with SMBG. In the SELFY prospective interventional single-arm study [4], which used the FreeStyle Libre system in 76 children and adolescents with T1D, mean HbA1c was reduced from 7.9% to 7.5% over 8 weeks, compared with SMBG. Separate prospective observational and randomized control studies show that flash glucose monitoring is associated with significant improvements in HbA1c in adults with T1D [5], or with T2D on insulin [6]. Moreover, meta-analysis of up to 25 real-world clinical studies has confirmed that starting the FreeStyle Libre system is associated with a significant and sustained reduction in HbA1c for adults and children with T1D [7,8] and for adults with T2D [7]. However, none of those studies correlated these improvements with the number of scans.

Flash glucose monitoring is now included in the Portfolio of Services of the Public Health System in Spain for T1D, and Andalucia has been a pioneer region in introducing flash glucose monitoring in the paediatric population in Spain. Here, we report on the impact of flash glucose monitoring over 6 months on measures of glycaemic control in a paediatric T1D population, being treated either with multiple daily injections of insulin (MDI), or with the continuous subcutaneous infusion of insulin (CSII).

The aim of the study is to understand the association between daily sensor scan rates and changes in HbA1c and in measures of hypoglycemia, in order to be able to establish from what number of scans per day an improvement can be seen, and from what point during the follow-up period (1 month, 3 months or 6 months) we will be able to give realistic recommendations to achieve the expected effect.

## 2. Methods

### 2.1. Study Design and Participants

We enrolled 145 pediatric T1D diabetes patients into a prospective, interventional study on the impact of the continuous use of FreeStyle Libre system on HbA1c levels, and on measures of hypoglycemia. The inclusion criteria were as follows: the presence of T1D with disease duration >1 year; an age of 4–18 years at study entry. Subjects were excluded if they did not adhere with the routine clinical review, or had used the FreeStyle Libre system prior to start of the study. Baseline characteristics are shown in Table 1. Mean age (±SD) of study subjects was 11.36 (±3.06) years, and the average duration of diabetes was 5.2 (±3.2) years. Patients were treated either with MDI (*n* = 119) or with CSII (*n* = 26).

All study subjects underwent group training, along with their main caregivers and diabetes team, to understand the functionality of the FreeStyle Libre flash glucose monitoring system, as well as the LibreView web platform which is used to view the glycaemic data collected by the FreeStyle Libre system.

The study was carried out in accordance with the regulations published in Boletín Oficial de la Junta de Andalucía (BOJA), with resolution on 17 April 2018, from the SAS management, for the inclusion of glucose monitoring systems in the Portfolio of Services of the Public Health System of Andalusia.

Data acquisition and analysis was performed in compliance with protocols approved by the Ethical Committee of the Provincial Ethics Committee of Malaga and Andalusian Ministry of Health and Family (ethical approval number: PIGE 0533-219). 

Written informed consent was obtained from all participants prior to study.

All data is available through contacting the authors.

### 2.2. Outcomes Measures

Laboratory measurement of HbA1c was carried out at baseline, and at 1, 3, and 6 months. Measures of hypoglycemia were monitored using the LibreView^®^ platform, (Abbott Diabetes Care, Witney, UK) using the most recent 14 days of glycaemic data at 1 month, 3 months and 6 months. Parameters measured included those accepted in international consensus guidelines for interpreting continuous glucose monitoring (CGM) data [9,10]. These were: the number of sensor scans per day; percentage of readings <70 mg/dL; number of Level 1 hypoglycemia events <70 mg/dL; number of Level 2 hypoglycemia events <54 mg/dL; percentage time in range (TIR) between 70–180 mg/dL. Across the study period, FreeStyle Libre users were divided into 4 groups based on their mean daily scan rates. These subgroups corresponded to the quartiles of the daily scan rates. These were: 0–6 scans/day; 7–8 scans/day; 9–11 scans/day; and >11 scans/day. Changes in reported glycemic measures were assessed in this context.

### 2.3. Statistical Analysis

Data were analysed using the established parametric method of comparing the means of normally distributed observations (paired Student’s t statistic), X^2^ for percentages. ANOVA was used for comparing different groups, as well as applying generalized mixed linear models, which can be more flexible in the analysis of data which incorporates multiple variables, such as metabolic control and insulin-treatment modality.

## 3. Results

### 3.1. Change in HbA1c in Relation to Daily Scan Rates with the FreeStyle Libre System

When we looked at the changes in HbA1c across the whole study group, there was a significant relationship between the change in HbA1c with the number of daily scans (*p* < 0.001), as demonstrated by the mixed linear modelling. Figure 1 shows that the change in HbA1c had a U-shaped relationship with the number of daily scans; HbA1c was higher at lower scan rates, and decreased as scan rate increased to between 15–20 scans, after which it raised at higher scan rates.

It is interesting to observe that from 15–20 scans a day glycosylated hemoglobin was negatively influenced. This fact may be related to an infective adherence, which shows repeated ineffective acts. These data had already been published with the number of capillary glycemic controls per day, where it was seen that a greater number of capillary glucose controls were related to better control measures in hemoglobin; however, from a certain number it was related to caregiver anxiety, and/or the patient, without having a favorable impact on metabolic control.

The confidence interval, represented as a dotted line in the graph, shows the dispersion of the results.

### 3.2. Relationship between Daily Scanning Rates and Change in Hypoglycemia

Hypoglycemia was assessed using three parameters based on FreeStyle Libre sensor data uploaded to the LibreView platform for the preceding 14 days at each study point. These were: percentage time with glucose readings <70 mg/dL; the number of clinically relevant Level 1 hypoglycemic events <70 mg/dL; the number of clinically significant Level 2 hypoglycemic events <54 mg/dL.

An analysis of change in hypoglycemia, based on the number of daily sensor scans, showed that there was a significant correlation between daily scan rates and the change in all three measures of hypoglycemia over the intervention period (Table 2). At each timepoint after starting FreeStyle Libre, there was a pattern of increasing % time <70 mg/dL, as well as an increase in events <70 mg/dL and <54 mg/dL, for groups with scan rates rising to 9–11 scans/day, then reducing for the group at >11 scans/day. With increasing time using the FreeStyle Libre system, the number of events <70 mg/dL and events <54 mg/dL decreased consistently, as scan rates increased. For the number of events <70 mg/dL, there was a significant reduction from 19.29 events/day to 12.69 events/day by 6 months (*p* < 0.001), for people with 9–11 scans/day, and from 13.57 events/day to 9.82 events/day (*p* = 0.01) for users with >11 scans/day. The number of events <54 mg/dL fell from 6.22 events/day to 3.68 events/day (*p* = 0.01) over 6 months for users with 7–8 scans/day, and from 7.50 events/day to 5.03 events/day (*p* = 0.04) for users with 9–11 scans/day. For % time <70 mg/dL, there was a significant fall from 5.8% to 3.88% (*p* = 0.023) from 1 to 6 months with scan rates of >11 scans/day.

In the variable number of events for <70 mg / dL in the 7–8 scan group, the difference at 6 months, compared with the third, was not significant nor clinically relevant (6.21 versus 6.52), so it cannot be taken into account. This scan group (7–8) only shows significant and concordant differences in the reduction of events of less than 54 mg/dL. For scan group of >11 scans/day there was a significant fell from 5.8% to 3.88% (*p* = 0.023) from 1 to 6 months, and for users with 9-11 scans/day fell from 7.50 events/day to 5.03 events/day (*p* = 0.04) at 6 months.

## 4. Discussion

This study, regarding a cohort of 145 children and young people aged 18 years or younger with T1D, shows that starting the FreeStyle Libre system is associated with improvements in glucose control for this group of people with diabetes. This includes reductions in HbA1c and improvements in hypoglycemia.

As they are part of other articles published in relation to glycosylated hemoglobin and the reduction of severe hypoglycemia, this article focuses on the role of wear time and adherence measured by the number of scans per day in metabolic control parameters.

This article has the purpose of being able to establish practical recommendations regarding the number of scans recommended to achieve the established objectives that do not enslave the patient or the caregiver, and allow realistic expectations for their use.

Moreover, we established that in case of more than 15–20 scans per day, an unfavorable impact was observed on the metabolic control valued in glycosylated hemoglobin. This may show an uneffective adherence and, it could be a diagnostic key of psycho-emotional exhaustion for those seeking positive results in the short and long term by raising the number of scans.

An important observation from our study is that there was a significant correlation between daily scan rates and the change in all three measures of hypoglycemia over the 6-month study period, including the number of hypoglycemic events < 54 mg/dL. Across the outcome timepoints after starting FreeStyle Libre, there was a pattern of decreasing % time < 70mg/dL, as well as a decrease in events <70 mg/dL and <54 mg/dL for groups with higher daily scan rates, which emphasizes again the importance of patient education and compliance with the use of the device, and acting according to the sensor readings revealed at higher scan rates. In the absence of masked baseline measures of hypoglycemia, the simplest explanation for this trend is that the different scan rates at month 1 are diagnostic of the level of glycemic control prior to starting the FreeStyle Libre. The lowest scanning group may correlate with those with poor prior control if they were previously low SMBG testers, and could likely have the least time with low glucose. Higher scan rates can indicate a desire for good control that is revealed by more events with glucose <70 mg/dL and <54 mg/dL in the early phase of using the FreeStyle Libre system.

However, as users become more experienced with the FreeStyle Libre system, at 3 months and at 6 months, the higher-scanning patients improve their glycemic performance compared to month 1, as they learn to use the full capabilities of the system. Thus, the number of low glucose events <70 mg/dL decreases significantly from month 1 to month 6, for the groups scanning 9 times per day or more, and the % time <70 mg/dL falls significantly for people with >11 scans/day. Since there was no masked baseline for measures of hypoglycemia, the interpretation for this might be that as scan rates increased, the FreeStyle Libre device provided diagnostic feedback on glucose levels, but above 11 scans/day the users were able to make therapeutic decisions themselves to avoid hypoglycemia.

Similarly, the number of clinically significant hypoglycemic events < 54 mg/dL fell significantly across the study period for people scanning between 7 and 11 times per day. Overall, this paints a picture of how the experience and engagement by the user makes an impact, such that they are able to interpret and act on their flash glucose data more effectively. By 6 months, the higher-scanning users were able to improve on their performance of the first month. This emphasises the value of the FreeStyle Libre system as a therapeutic tool in terms of reducing hypoglycemia at higher scan rates for experienced users. This observation is aligned with a real-world analysis of associations between FreeStyle Libre sensor-scanning frequency and a range of glycaemic measures [11].

These observations are aligned with data from the AWeSoMe real-world study [12] on 71 young people, aged from 1–25 years old with T1D, who had self-funded their FreeStyle Libre system; however, this latter study did not include a masked baseline set of readings. The SELFY single-arm study in 76 children and young people did not show a significant change in % time in hypoglycemia <70 mg/dL, despite a 14-day masked baseline period of wear [4], which appeared to differ from the overall pattern that we saw in our study. However, the SELFY study was an 8-week multicentre study, compared with our single-centre 6-month real-world study, and showed a significant reduction in the number of events <70 mg/dL, in common with our observations.

Reduction in the risk of severe hypoglycemia amongst children and young people with T1D has been reported in only one other study to date, which showed a 53% reduction in the rate of severe hypoglycemia for 278 subjects after 12 months, following a switch from SMBG testing to the use of the FreeStyle Libre system [13].

Although the reduction in hypoglycemia is present in other studies on monitoring systems, ref. [12] the Flash device has one peculiarity in comparison with other monitoring devices: it is easier to manage, because it is not necessary to calibrate it.

Regarding HbA1c, it is curious that after 20 scans HbA1c increased in our sample. This could be a result of ineffective repeated impulsive behaviors. These data had already been published with the number of capillary glycemic controls per day, where it was seen that a greater number of capillary glucose controls were related to better control measures of hemoglobin [12]; however, from a certain number, this was related to caregiver anxiety, and/or the patient, without having a favorable impact on metabolic control.

A limitation of our study is that the longitudinal changes in measures of hypoglycemia were not able to include a baseline reading for those derived from the masked use of the FreeStyle Libre system, prior to users becoming aware of their sensor-glucose readings. Thus, changes in these metrics across the 6-month study period can be argued to be a study effect, as the consequence of the users becoming better educated in the day-to-day management of their diabetes was due to their participation in the study.

## 5. Conclusions

This prospective observational report from a single centre in Spain underlines both the value of flash glucose monitoring with the FreeStyle Libre system, and also the importance of understanding individual glycaemic profiles. In our study, we show that adherence with the FreeStyle Libre system is important to fully realize the benefits of flash glucose monitoring. Our study also indicates that additional investigation is required to identify which children and young people with T1D are most likely to benefit from use of the FreeStyle Libre system.

## Figures and Tables

**Figure 1 jcm-11-00286-f001:**
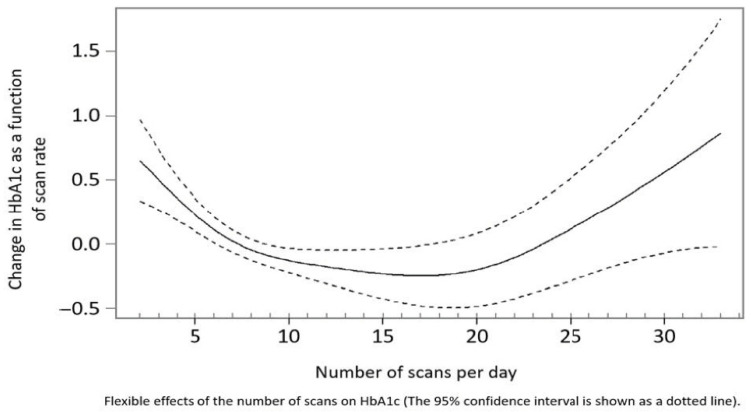
Relationship between number of daily scans with the FreeStyle Libre system and HbA1c.

**Table 1 jcm-11-00286-t001:** Baseline clinical characteristics of the study population.

Parameter		*n*
Age (years)	11.36 ± 3.06	145
Duration of diabetes (years)	5.2 ± 3.2	145
Mean HbA1c at baseline (%)	7.11 ± 0.80	142
Mean HbA1c for subjects <7.5% at baseline	6.73 ± 0.47	102
Mean HbA1c for subjects ≥7.5% at baseline	8.09 ± 0.71	40
Treatment type	MDI	119
Treatment type	CSII	26

Values are mean ± SD unless otherwise indicated. MDI, multiple daily injections of insulin; CSII, continuous subcutaneous insulin infusion.

**Table 2 jcm-11-00286-t002:** Relationship between daily scanning rates and change in hypoglycemia.

% Time <70 mg/dL (±SE)				
No daily scans	Month 1	Month 3	Month 6	*p* value *
0–6	4.75 (0.55)	4.94 (0.37)	4.83 (0.41)	0.909
7–8	6.64 (0.69)	5.48 (0.51)	5.26 (0.41)	0.169
9–11	7.92 (0.81)	6.21 (0.47)	6.52 (0.47)	0.14
>11	5.89 (0.81)	4.53 (0.39)	3.88 (0.34)	0.03
Number of events <70 mg/dL (±SE)				
No daily scans	Month 1	Month 3	Month 6	*p* value *
0–6	8.22 (0.96)	7.74 (0.50)	5.97 (0.45)	0.06
7–8	13.30 (1.15)	9.81 (0.68)	11.16 (0.60)	0.1
9–11	19.29 (1.66)	13.27 (0.71)	12.69 (0.66)	<0.001
>11	13.57 (1.34)	12.37 (0.68)	9.82 (0.55)	0.01
Number of events <54 mg/dL (±SE)				
No daily scans	Month 1	Month 3	Month 6	*p* value *
0–6	1.93 (0.37)	2.73 (0.29)	2.28 (0.28)	0.455
7–8	6.22 (0.83)	3.00 (0.38)	3.68 (0.34)	0.01
9–11	7.50 (1.12)	4.54 (0.42)	5.03 (0.42)	0.04
>11	3.38 (0.65)	3.45 (0.35)	3.28 (0.32)	0.897

Data are means (±standard error) in each case. * *p* value for change in readings at 6 months compared to 1 month.

## Data Availability

All data is available through contacting the authors.

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
