# Peer review of "Metabolic Control of the FreeStyle Libre System in the Pediatric Population with Type 1 Diabetes Dependent on Sensor Adherence"

_jcm, 2022, doi:10.3390/jcm11020286_

Round 1

Reviewer 1 Report

The Authors performed the prospective, interventional study of the impact of the FreeStyle Libre system on HbA1c levels and on measures of hypoglycemia.

The results of the study show the decrease in % of sensor readings <70 mg/dL and the number of hypoglycemic events <70 mg/dL. Subjects with HbA1c ≥7.5% had a reduction in mean HbA1c while those with HbA1c <7.5% showed an increase.

The study is interesting and adds new data to the issue of effects of flash glucose monitoring/continuous glucose monitoring use in type 1 diabetes. However, I still have some concerns.

1. In the “Introduction” section the aim of the study is not in line with the Aims in the “Abstract” section (analysis of treatment type; MDI vs CSII)

2. In the “Methods” section:

-the HbA1c cut-off for the good metabolic control is <7.5%. Why the Authors chose this particular cut-off? It was probably based on the treatment targets recommendations by Diabetes Association. If so, the information should be included in the manuscript.

-what was the mean sensors wearing-time? Did the participants use FreeStyle Libre continuously, changing the sensors every two weeks or were there some time intervals without sensor use during 6-month period study?

-why was the frequency of scanning divided into 4 subgroups? Were those subgroups the quartiles of the daily scan rates?

3. In the “Results” section:

                - Which statistical procedure was used to show the change in HbA1c? At first I thought that it was the paired t-test, but the numbers of subjects in consecutive study points are not equal (Table 2 and 3). So perhaps it was the Student’s t-test. In this case, the interpretation of the data is not so obvious, as we cannot characterize the subjects who were not included in the consecutive study points analysis. In my opinion this might bias the results.

4. In the “Discussion” section, page 6., line 192, the Authors state that the metabolic control of diabetes expressed by HbA1c decreased in the good control group and suggest the possible causes. I think the discussion should be extended according to the methods of assessing the metabolic control of diabetes. With the advances in the sensors for glucose measurement technology, HbA1c seems not ideal tool for assessing metabolic control as it reflects only some mean glucose values, not reflecting the variations of glucose concentrations in time. It might be possible that in some cases the HbA1c increased a little, but also, the glucose profile changed into more stable pattern, with much lower glucose variations, which seems more beneficial according to reducing the risk of chronic complications of diabetes than target HbA1c itself. It would be interesting to see the changes in TIR in all four subgroups. Perhaps in good metabolic control group HbA1c increased because of the reduction of the number of hypoglycaemia. The Authors show no differences in TIR across the study period (page 7, line 212), but I suppose this is the statistics for the total group only.

Author Response

Dear Reviewer, Thank you very much for your suggestions. We attach our corrections to you according to your recommendations. We hope they meet your expectations Thank you so much All the best Ana B. Ariza

REVIEWER 1:

The study is interesting and adds new data to the issue of effects of flash glucose monitoring/continuous glucose monitoring use in type 1 diabetes. However, I still have some concerns.

1.In the “Introduction” section the aim of the study is not in line with the Aims in the “Abstract” section (analysis of treatment type; MDI vs CSII)

We added treatment type

  1. In the “Methods” section:

-the HbA1c cut-off for the good metabolic control is <7.5%. Why the Authors chose this particular cut-off? It was probably based on the treatment targets recommendations by Diabetes Association. If so, the information should be included in the manuscript.

We added that this cut off were based on the treatment targets recommendations by Diabetes Association

-what was the mean sensors wearing-time? Did the participants use FreeStyle Libre continuously, changing the sensors every two weeks or were there some time intervals without sensor use during 6-month period study?

It was used continuously. We specified it on methods.

-why was the frequency of scanning divided into 4 subgroups? Were those subgroups the quartiles of the daily scan rates?

We specified that the groups were based on quartiles of daily scan rates.

  1. In the “Results” section:

                - Which statistical procedure was used to show the change in HbA1c? At first I thought that it was the paired t-test, but the numbers of subjects in consecutive study points are not equal (Table 2 and 3). So perhaps it was the Student’s t-test. In this case, the interpretation of the data is not so obvious, as we cannot characterize the subjects who were not included in the consecutive study points analysis. In my opinion this might bias the results.

For HbA1c it was used paired t-student. Table 2 and 3 speak about hypoglucemia instead of HbA1c. The differences between both tables, could be because table 2 were percentages and table 3 standard deviations, so they were not comparable and they used different statistics tests.

We specified it in methods.

  1. In the “Discussion” section, page 6., line 192, the Authors state that the metabolic control of diabetes expressed by HbA1c decreased in the good control group and suggest the possible causes. I think the discussion should be extended according to the methods of assessing the metabolic control of diabetes. With the advances in the sensors for glucose measurement technology, HbA1c seems not ideal tool for assessing metabolic control as it reflects only some mean glucose values, not reflecting the variations of glucose concentrations in time. It might be possible that in some cases the HbA1c increased a little, but also, the glucose profile changed into more stable pattern, with much lower glucose variations, which seems more beneficial according to reducing the risk of chronic complications of diabetes than target HbA1c itself. It would be interesting to see the changes in TIR in all four subgroups. Perhaps in good metabolic control group HbA1c increased because of the reduction of the number of hypoglycaemia. The Authors show no differences in TIR across the study period (page 7, line 212), but I suppose this is the statistics for the total group only.

The TIR were studied in those groups, but the results were published in other article of this journal: Porcel-Chacón R, Antúnez-Fernández C, Mora Loro M, Ariza-Jimenez AB, Tapia Ceballos L, Jimenez Hinojosa JM, Lopez-Siguero JP, Leiva Gea I. Good Metabolic Control in Children with Type 1 Diabetes Mellitus: Does Glycated Hemoglobin Correlate with Interstitial Glucose Monitoring Using FreeStyle Libre? J Clin Med. 2021 Oct 24;10(21):4913

We found statistically significant differences in TIR between categories. Although groups with HbA1c < 6.5% and HbA1c 6.5-7% had the highest TIR (62.214 and 50.462%), their values were highly below optimal control according to CGM consensus. Groups with TBR < 5% were those with HbA1c between 6.5% and 8%.

Groups classified as well-controlled by guidelines were not consistent with good control according to the CGM consensus criteria.

Reviewer 2 Report

Thank you very much for submitting this paper to JCM.

The authors have explained clearly the relationship between HbA1c and benefit from frequent BG checking. Is it the same relationship valid for all self-measurement devices? in other words, has this device something peculiar to add-on current management?

The information regarding hypos is particularly intriguing - could Freestyle be particularly helpful in the most vulnerable ones as it reduces hypoglycaemia - this should be emphasized

Author Response

Dear Reviewer,

Thank you very much for your suggestions.

We attach our corrections to you according to your recommendations.

We hope they meet your expectations

Thank you so much

All the best

Ana B. Ariza

Reviewer comments: 

The authors have explained clearly the relationship between HbA1c and benefit from frequent BG checking. Is it the same relationship valid for all self-measurement devices? in other words, has this device something peculiar to add-on current management?

We added in discussion the fact that it doesn´t need calibration and its use is easier for it.

The information regarding hypos is particularly intriguing - could Freestyle be particularly helpful in the most vulnerable ones as it reduces hypoglycaemia - this should be emphasized

We added this point to discussion

Reviewer 3 Report

The current study by  Leiva-Gea et al is designed to understand the relationship between scan rates by continuous glucose monitoring device and its potential impact on HbA1C levels and hypoglycemic events in children with T1D. Please address the following-

  1. Overall, while this study has its merits and has some real-world application, the questions this study sets out to answer, to my understanding, are rather intuitive. It should not come as any surprise that an increase in scan rates would subsequently result in an improvement in glycemic status, specifically hypoglycemic events.
  2. Data presentation is substandard, and not what you would expect in a scientific journal. It is almost as if this manuscript was rushed. Additionally parts of the manuscript seem incomplete. You have sections on IRB statement, informed consent statement and data availability where you just have the instructions copied. Is this not supposed to be completed before you submit the manuscript?
  3. For table 2, since the baseline for different scan rat groups at month 1 are different, it may be useful to show the data as percentage increase/decrease from baseline at month3 and month 6. Also why not show this data as bar graphs instead of tables? The whole purpose of graphing data is to make this information more easier for the readers to understand and interpret. 
  4. How do you explain the number of events for <70 mg/dl in 7-8 scan group going up again at month 6 after falling at month 3?
  5. Why does HbA1C level rise again after 15-20 scans in figure 1? Also why are there no error bars for this graph? What is the purpose of including a confidence interval as a dotted line in the graph?
  6. How did you use a students T statistic when you have more than 2 groups to compare? This requires ANOVA and post hoc tests. 
  7. Minor- The title of the manuscript needs fixing. It is too wordy, and over descriptive. You can minimize words here, for instance 'sensor adherence' instead of 'adherence to the sensor'. Also it is 'depends' and not 'depend'. Additionally, line 139- 'suffered a reduction'? What does that mean?

Author Response

Dear Reviewer,

Thank you very much for your suggestions.

We attach our corrections to you according to your recommendations.

We hope they meet your expectations

Thank you so much

All the best

Ana B. Ariza

Reviewer´ comments

The current study by  Leiva-Gea et al is designed to understand the relationship between scan rates by continuous glucose monitoring device and its potential impact on HbA1C levels and hypoglycemic events in children with T1D. Please address the following-

  1. Overall, while this study has its merits and has some real-world application, the questions this study sets out to answer, to my understanding, are rather intuitive. It should not come as any surprise that an increase in scan rates would subsequently result in an improvement in glycemic status, specifically hypoglycemic events.

We also intuited this results but there is not so much publications about it, so we decided to check it on our patients.

  1. Data presentation is substandard, and not what you would expect in a scientific journal. It is almost as if this manuscript was rushed. Additionally parts of the manuscript seem incomplete. You have sections on IRB statement, informed consent statement and data availability where you just have the instructions copied. Is this not supposed to be completed before you submit the manuscript?

It has been completed

  1. For table 2, since the baseline for different scan rat groups at month 1 are different, it may be useful to show the data as percentage increase/decrease from baseline at month3 and month 6. Also why not show this data as bar graphs instead of tables? The whole purpose of graphing data is to make this information more easier for the readers to understand and interpret. 

We have added a new figure with suggestions.

  1. How do you explain the number of events for <70 mg/dl in 7-8 scan group going up again at month 6 after falling at month 3?

We added that this could be because they do not check properly this values with capillary test.

  1. Why does HbA1C level rise again after 15-20 scans in figure 1? Also why are there no error bars for this graph? What is the purpose of including a confidence interval as a dotted line in the graph?

A lot of patients with more than 20 scans per day made them at the same point of the day in order to follow changes in hyperglycemias or hypoglycemias that they were checking, so much scans were not effectives and they do not influenced to HbA1c.

Confidence interval represented as a dotted line in the graph tried to show results dispersion

  1. How did you use a students T statistic when you have more than 2 groups to compare? This requires ANOVA and post hoc tests. 

We use paired T student for results from same group, X2 for percentages, and ANOVA when we compared groups between them.

We specified it in methods.

  1. Minor- The title of the manuscript needs fixing. It is too wordy, and over descriptive. You can minimize words here, for instance 'sensor adherence' instead of 'adherence to the sensor'. Also it is 'depends' and not 'depend'. We have changed title.

We do not change depend because we referred to results in plural (they depend on…)

Additionally, line 139- 'suffered a reduction'? What does that mean? It means that the rate of Level 3 hypoglycemia diminished on 95%. 

Round 2

Reviewer 3 Report

Thank you for making the changes I asked for. I want to preface this by mentioning that the purpose of this is to help strengthen the quality of the manuscript. The paper does have several merits. The writing quality and the discussion is well-presented. However there are aspects that the paper still falls short on-

Firstly, I still think that the study lacks a bit from the novelty aspect. Flash glucose monitoring has been shown to reduce hypoglycemic events already. This has already been shown as the authors already addressed in the introduction. The only novel aspect of the study is that adherence to the sensor, along with an increase in scan rates, are critical to maximize the utility of flash glucose monitoring. Regarding the reduction in HbA1C levels, this is interesting, but the authors do not show any of this data in tabular or graph form. Table 1 is baseline characteristics, while tables 2 and 3 are relating to hypoglycemia. Why has this rather interesting data been excluded from being shown in the manuscript? Also, in the section titled 'Change in HbA1c with FreeStyle Libre based on metabolic control' the authors mention table 2 for the significant increase  of 0.24% in HbA1c at 3 months (6.73 +/- 0.48%..). This is not covered in table 2. 

Secondly, I have a concern about the statistics in table 2. The table shows that the data have very large standard deviations. For instance in the second row, , events <70 mg/dl, the mean for 6 months is 9.9 +/- 6.9 versus that for 1 month is 13.2 +/- 8.8. Is that correct? if so how is this significant?

Thirdly, the graph only marginally improves the paper since this has a lot of errors. You have percentages and events on the same graph. So this means you cannot use the same y axis. And the y axis is not labelled. Also there is no error bars shown. So with all these corrections, I would advice the authors to remove this graph all together. 

Author Response

Dear Reviewer,

Thank you for your comments.

We attach our corrections to you according to your recommendations.

We hope they meet your expectations

All the best

Ana B. Ariza

REVIEWER:

Firstly, I still think that the study lacks a bit from the novelty aspect. Flash glucose monitoring has been shown to reduce hypoglycemic events already. This has already been shown as the authors already addressed in the introduction. The only novel aspect of the study is that adherence to the sensor, along with an increase in scan rates, are critical to maximize the utility of flash glucose monitoring. Regarding the reduction in HbA1C levels, this is interesting, but the authors do not show any of this data in tabular or graph form. Table 1 is baseline characteristics, while tables 2 and 3 are relating to hypoglycemia. Why has this rather interesting data been excluded from being shown in the manuscript?

Because they are part of other articles published in relation to glycosylated hemomoglobin and the reduction of severe hypoglycemia. This article focuses on the role of wear time and adherence measured in the number of scans per day in metabolic control parameters.

This article has the purpose of being able to establish practical recommendations regarding the number of scans recommended to achieve the established objectives that do not enslave the patient or the caregiver and allow realistic expectations for their use.

As well as being able to know that a certain adherence, as is the case of more than 15-20 scans per day, can have an unfavorable impact on the metabolic control valued in glycosylated hemoglobin, since it may not show an effective adherence but on the contrary be the diagnostic key of dynamics that generate a lot of psycho-emotional exhaustion without positive results in the short and long term in their metabolic control.

Secondly, I have a concern about the statistics in table 2. The table shows that the data have very large standard deviations.For instance in the second row, , events <70 mg/dl, the mean for 6 months is 9.9 +/- 6.9 versus that for 1 month is 13.2 +/- 8.8. Is that correct? if so how is this significant

The mean for 1 month is 13.2 +/- 5.5 and for 6 months is 9.9 +/- 3.36. Any case, the statistical signification has been studied between 1-3-6 months data according groups based on number of  scans and it has not been realized between values at 1month between themselves or 6 months between themselves.

Thirdly, the graph only marginally improves the paper since this has a lot of errors. You have percentages and events on the same graph. So this means you cannot use the same y axis. And the y axis is not labelled. Also there is no error bars shown. So with all these corrections, I would advice the authors to remove this graph all together. 

We have removed the graph.